# Emergence of Organisms

**DOI:** 10.3390/e22101163

**Published:** 2020-10-16

**Authors:** Andrea Roli, Stuart A. Kauffman

**Affiliations:** 1Department of Computer Science and Engineering, Alma Mater Studiorum Università di Bologna, Campus of Cesena, I-47522 Cesena, Italy; 2European Centre for Living Technology, I-30123 Venezia, Italy; 3Institute for Systems Biology, Seattle, WA 98109, USA; stukauffman@gmail.com

**Keywords:** emergence, evolution, critical dynamics, cybernetics, information, biosemiotics, constraint closure, Kantian whole, affordance, entailing laws, consciousness

## Abstract

Since early cybernetics studies by Wiener, Pask, and Ashby, the properties of living systems are subject to deep investigations. The goals of this endeavour are both understanding and building: abstract models and general principles are sought for describing organisms, their dynamics and their ability to produce adaptive behavior. This research has achieved prominent results in fields such as artificial intelligence and artificial life. For example, today we have robots capable of exploring hostile environments with high level of self-sufficiency, planning capabilities and able to learn. Nevertheless, the discrepancy between the emergence and evolution of life and artificial systems is still huge. In this paper, we identify the fundamental elements that characterize the evolution of the biosphere and open-ended evolution, and we illustrate their implications for the evolution of artificial systems. Subsequently, we discuss the most relevant issues and questions that this viewpoint poses both for biological and artificial systems.

## 1. Introduction

Can one formulate broad principles for evolving proto-cells, single cell organisms, perhaps multi-celled organisms and even robots, which live in their complex worlds, adapt, and survive, can grow more complex and diverse in an abiotic or biotic environment co-evolving with one another? Building on Ashby’s ideas presented in *Design for a brain* [1], we address these questions trying to identify a minimal set of necessary and sufficient properties that are likely to characterize living organisms—and also artificial systems—that evolve in open environments.

A first property is the ability of discriminating what is beneficial or disadvantageous for the organism, "what’s good or bad", in Ashby’s terms. This discrimination capability consists in being able to classify relevant information from the environment, and it leans on sensors that can capture the relevant information for categorizing external stimuli. The detection of relevant information is a way of meaning creation: what is important for the survival of the organism in its environmental niche shapes the evolution of specialized sensors, so the patterns and the correlations they capture come to exist and get a name. Not just data streams collected by sensors, but semantic information. Besides the capability of capturing relevant information from the environment, for surviving the organism must be able to use this information properly. As a consequence, mechanisms and means for acting in the world should also be developed. In robotic terms, we may speak of actuators and effectors [2], but we also include here the ability of taking decisions and act, i.e., having a control policy—that should also be adapted and changed.

The presence of these properties in an evolutionary setting that is characterized by heritable variations and selection, along with individual adapting and learning mechanisms, is the basis for enabling open-ended evolution. We observe that these capabilities are common to all living organisms, but they may well be crucial also for proto-cells. For a proto-cell, sensing its world, evaluating it as “good or bad for me” and acting reliably and appropriately would have been of enormous selective advantage.

In this work, we first discuss the limits of current approaches to artificial evolution and adaptation in Section 2, by outlining the elements that are relevant for our discussion with particular focus on robotics. In Section 3, building upon Ashby’s argument, we detail the properties that organisms and possibly artificial systems should possess to be able to exhibit adaptive behavior and evolve. The implications of these hypotheses are discussed in Section 4 and Section 5, where we elaborate on affordances, constraint and work closures, semantic information, and dynamical criticality, which play an essential part in open evolution and co-evolution of systems. Finally, in Section 6, we debate whether mind has some role in the evolution of the biosphere.

## 2. Current Limits in Robotic Evolution and Adaptivity

Robot designers usually try to endow their creatures with some kind of adaptivity, which can range from the simple automatic adjustment of a proximity sensor threshold to the capability of learning new plans to achieve given goals [3]. This property enables the robot to face perturbations in an unpredictable environment, keep its basic functionalities (i.e., self-sufficiency), and pursue its goals. Early cyberneticists, such as Ashby [1], refer to the notion of homeostasis, meaning that the robot is able to keep some essential variables within given limits. We can generalize the notion of homeostasis adding to self-sufficiency also the requirement to achieve given goals (a goal can be “reach the light”, “sweep the floor”, but also more generic, such as “explore new areas”). Every kind of adaptive mechanism involves some modification inside the robot, either in its morphology or, more often, in the controller.

Among the kinds of adaptivity, biologists also include evolution, which acts upon populations of individuals and adapt them by genetic adjustments. Artificial evolution is also used to design robots [4,5]. The principles of Evolutionary Robotics (ER) are rather simple: the controller (and sometimes also the morphology) of a robot is encoded in a formal representation subject to genetic-like operations, such as mutation and recombination. The fitness of a robot is estimated by evaluating robot’s behavior, almost always in a simulator. Subsequently, on the basis of fitness assessment, selection takes place and new individuals are produced by applying genetic operators. This cycle is iterated until a given termination condition is met, e.g., a solution with sufficiently high fitness has been found. Evolutionary robotics has attained interesting results, especially concerning proofs of principle, but it is still not particularly successful in applications [5,6]. Setting aside the questions concerning possible improvement of ER as an engineering tool, we emphasize two important points. First, ER almost always depends upon a simulator; this means that fitness is evaluated in a different environment than the one in which the robot will operate and, above all, the simulator is supposed to reproduce at least all the properties of the world that the designer believes are relevant for the robot to accomplish its task. (There might be also non-relevant properties, of course, that the evolutionary process might anyway exploit) Second, the fitness function is externally defined and it should not just represent a coherent evaluation of robot’s behavior, but it should also provide a guide to the evolutionary process (i.e., it should be defined in such a way that the fitness values and genotype distance in terms of genetic operators are sufficiently correlated). These two latter observations also hold for most adaptive mechanisms, not based on artificial evolution.

In general, we can state that, when we build robots, we prescribe in advance: evaluation function (i.e., the robot is programmed already to know what’s good for it), sensors (at least the relevant ones), actuators (at least the ones needed for making actions according to the maximization of the evaluation function). In short, we have constructed the robot such that it can learn to do what we want to do within the world that it can know and in which it can act with respect to the relevant features we have predefined (notice that evolution of life predefines none of these).

If nothing changes in the robot and in the environment, at the end of the process we can find a robot able to attain sufficiently high values of the evaluation function. In this setting, nothing actually novel can emerge, as, in principle, boundary and initial conditions are known.

It is important to emphasize an issue concerning simulation: the simulator is an abstraction of the world encoded in a formal language; therefore, deduction can, in principle, be applied to entail the possible outcomes of an artificial evolutionary process [7]. Here, we exclude phenomena due to bugs in the code that can be exploited by the evolutionary process (e.g., for an error in the implementation of the energy consumption rule, moving backward increases robot’s energy). These are unexpected behaviors due to incomplete knowledge of the designer about the simulator. The point is that usually in robotics simulations there is no concept or idea that a physical object can be used in more than one way. Using objects in more than one way is a source of emergent behavior. Let’s consider the many uses of a screwdriver: it is not possible to conceive an algorithm that can either list all the possible uses of the screwdriver, or enumerate them implicitly by returning the "next use". The number of uses of a screwdriver is not infinite, but rather indefinite (the uses are in a nominal scale). Conversely, in a simulator, all of the elementary properties of objects and their relations with other objects are modeled; therefore, in principle, we could list or define a procedure to enumerate all the possible uses of an object, hence no actual emergent behavior can take place.

Anyway, in the limits of the initial and boundary conditions provided by the simulator, some sort of emergent phenomena can still arise. For example, simulated robots may develop signaling behaviors that are beneficial to group evolution [8], or they may find a sensory-motor coordination pattern that solves a complicated task [4]. These behaviors might be unexpected, as many phenomena observed in artificial evolution and simulation are [9]. Nevertheless, these behaviors are limited by the boundary conditions defined by the simulator and, therefore, the phase space in which the evolution of the robots takes place does not change. Hence, while simulation is useful for practical purposes, for observing actual and open emergent behaviors, we need robots embodied in the physical world.

However, embodiment alone is not enough. Let us suppose to set up the adaptive process, either evolutionary or not, in the physical world so as to overcome the limitations of simulation. Additionally, in this case, all the properties of the world and the robots are anyway pre-specified by the designer, starting from sensors and actuators. For example, if the task the robot has to accomplish is to sweep the floor in a room, autonomously going to the recharging area when necessary, practical constructive reasons would suggest the designer to equip the robot only with the necessary sensors (e.g., battery level indicator, proximity sensors, and camera) and actuators (e.g., wheels, recharging plug, vacuum cleaner). It would be quite improbable to find a designer mounting also a thermometer and a laser pointer. The reason is that the designer, often forced to satisfy constructive and energetic requirements, has conceived in advance that a solution (in fact, one of the many ones) will make use of those specific sensors and actuators. The adaptive process will then produce a suitable behavior policy for this specific context. While this approach is a perfectly plausible way for practical purposes, it is inappropriate when robots are the setting for constructive biology or cybernetic experiments, because this is surely not representative of biological adaptation phenomena. Therefore, the question arises as to what are the choices we have to take if we want to set up the conditions for the actual emergence of organisms, following Ashby who asked “to what extent is the machine restricted by the limitations of its designer?” [10].

## 3. Towards the Design for an Evolving Organism

In this section, we discuss the elements that we believe are at the core of the emergence of organisms, either natural or artificial. As we will see, these ingredients enable the properties that characterize open-ended evolution in the biosphere and, therefore, provide a viable way for addressing emergence and evolution in the artificial world.

### 3.1. The Evaluation Function

In robotics, adaptive mechanisms, such as learning and evolution, require the definition of a merit factor that is used as a feedback for the adaptive process, e.g., a fitness function in evolutionary computation techniques. These functions are externally defined, but their implicit purpose is to provide a value system to the robot, such that it is able to take decisions that are beneficial to its self-sufficiency and goals. Living organisms act so as to take what is good for their survival and goal achievement. On the other hand, by acting in this way they are likely to survive longer and, thus, spread their genetic material to the next generations. In essence, if they are endowed with a suitable value system and act accordingly, they attain both an individual advantage, as they keep homeostasis and achieve their goals, and a phylogenetic one. Let’s start from the simplest condition: a bacterium must find food to survive, hence, if the actions it takes lead it to the food, then they are "good", otherwise they are labeled as "bad". This binary choice becomes more articulated when quantities also come into play: the bacterium chooses the way to more food, over the one with less food. Therefore, this value is no longer binary, but ranging across a continuum. In the general case, organisms face conflicts among possible options and must make a choice, and their value system provides the basis upon which to choose. In this context, emotions have a primary role, as they are involved in the self-regulatory sensory system of organisms and category formation [11].

Ashby [1] suggests formalizing this concept by considering essential variables: if essential variables are kept in a given range, then the system is fine, i.e., if the system is able to act, so that its essential variables are within the given range, then it survives. This notion can be easily generalized to include in survival not just self-sustainability, but also the achievement of goals [12]. A dynamical systems perspective of agents and environment [13] provides a suitable formal framework for this generalization.

From an abstract viewpoint, we can state that surviving means keeping organismal individuality, i.e., the property enjoyed by Kantian wholes that achieve constraint closures and self-reproduce (a Kantian whole is an organized being that has the property that the parts exist for and by means of the whole [14,15]). These systems achieve catalytic closure: "each reaction or non-equilibrium process which must be catalyzed finds a catalyst in the system itself" [16]. An individual is a Kantian whole that achieves constraint closure, whereby each of a closed set of non-equilibrium processes constructs constraints that enable other processes to construct further constraints, in a circular way [17,18,19]. These Kantian wholes are also capable of heritable variations and natural selection, such that they can co-evolve to be critical. Constraints enable a process to do work, which is the constrained release of energy into a few degrees of freedom [20,21]. Constraint closure, plus the other things that constraints closed systems can build, define the boundaries of the individual. The individual lives in its abiotic and biotic world, defining its niche. The niche of an individual is what von Uexküll calls *Umwelt* [22,23], i.e., the subjective world of an organism, and it cannot be defined non-circularly. A further property is that organisms should be mutually critical. While the notion of criticality will be discussed in length in Section 5, we briefly anticipate here that organisms should be mutually critical, because they need to understand and interact with each other [24]. Finally, we observe that there have been recent attempts to provide rigorous quantitative methods for identifying individuals based on information theory [25], which emphasizes the role of the capability of using past information to condition future actions. In addition, a hybrid view that integrates the historical and relational conception of identity [26] has recently been proposed [27].

### 3.2. Sensors and Actuators

As observed in Section 2, if robots are equipped ab initio with sensors that acquire the information that is necessary for their survival, then the boundary conditions for their adaptation and evolution are already set and cannot change. Therefore, we need to set up a mechanism enabling the organism to extract and/or create useful information from the environment in order to avoid to inject ad hoc knowledge. To this purpose, we need either an evolutionary mechanism acting on populations of organisms (primarily by heritable variation and selection) or providing the organisms the capability of adaptively evolving and/or constructing sensors, or both. Cybernetics scholars have indeed explored the evolution of sensors, starting from the pioneer work by Pask [28] to more recent works [29,30,31]. As remarked by Cariani in [32] rephrasing Ashby, “in order to achieve better performance over its initial specification, a device must be informationally open, capable of interacting with the world independently of its designer, the device must have some degree of epistemic autonomy in order to improve itself, but epistemic autonomy is not achievable without some degree of structural autonomy”. In other terms, no improvement is possible if the organism is not able to change something in its structure, so as to autonomously find ways to profit from useful correlations in its environment. For example, if energy sockets are under light bulbs, a robot could detect and reach them faster if it is capable of developing an electromagnetic sensor for the visible spectrum. In biology, we have plenty of notable examples of sensor evolution, from the capability of bacteria of measuring food concentration [33], to the various forms of eyes [34].

Sensors are the way that organisms and robots acquire useful information that is used to act in the world through actuators. We assume a wide definition of actuator as a tool or mechanism that the robot can control to change something in the world (e.g., grasping an object, moving, lighting up a LED). Sensors and actuators are the information channels between the organism and environment and their combined use produces what in robotics is called sensory-motor loop [35], referring to the fact that sensors readings affect actuator commands and conversely the results of actions affect sensor readings (note that, while in robotics the notion of sensory-motor loop is based on the relation between the robot and the environment, in general a sensory-motor pattern in an organism does not require the definition of an “outside” and can be only internal to the system). Therefore, it is no surprise that the evolution of sensors and actuators is intertwined in Nature. Hence, in an artificial setting we also need to set up a mechanism enabling the organism to act properly and choose what is good for it. Again, we need either an evolutionary mechanism acting on populations of organisms or providing the organisms the capability of adaptively constructing actuators, or both. *En passant*, we observe that, while the subject of sensor evolution has received attention from the cybernetic community, the evolution of actuators has not been discussed so in depth [29]. We believe the reason is to be found in the fact that, rather than actuator evolution, the interest has been often focused on tool development.

Sensor and actuator construction essentially consists in identifying affordances, so as to make use of something that is “useful to me”. Informally speaking, an affordance is a feature of the world that can be used to do something useful [15,36]. This term has been introduced by Gibson [37] to capture the fact that objects afford observers possible actions, which are directly perceived. Jamone and co-authors [38] emphasize two aspects of affordances that are relevant for our discussion:affordances are not properties of the environment alone, but they depend on sensing and actuating capabilities of the robot; and,affordance perception suggests action possibilities to the robot through the activation of sensory-motor patterns, and it also provides a mean to predict the consequences of actions.

As a consequence, without the possibility of evolving its own sensors and actuators, a robot cannot identify affordances and so there are no ways for it to explore possible information acquisition and actions, and so improve. Note that an affordance may also consist in cooperating with or exploiting other organisms’ features. An affordance can be identified either because it brings an advantage to the organism (i.e., an accidental event that turns out to be advantageous [21]), or because it opens up an unexpected and favorable possibility (e.g., a particular sensory-motor pattern enables the robot to detect the size of an object). Furthermore, the fact that a given object or situation affords an organism to do something is also the way that things get meaning. The relation between affordances and meanings is central to our argument and it will be discussed in Section 4.

### 3.3. The Controller

The missing piece in the design of an organism is a mechanism for converting the information that is sensed by the robot, and possibly its state, into actions (i.e., the “controller” or the “control software”). Metaphorically, we can say that the system needs a behavior policy that maps perceptions and internal states to actions. If states and actions can be formally modeled and are time invariant, then any formalism defining a policy is sufficient for providing a functioning controller (e.g., a Markov machine), to be trained by a learning technique. Nevertheless, here, we are considering the case in which sensors and actuators can evolve in time, so both the sets of states and actions can change, and new states should be added to the policy and also new actions. In general, acting properly requires dynamics and choice. Therefore, a viable formalism for accommodating such requirements is that of dynamical systems, provided that they can be subject to structural changes (e.g., new variables can be added). Memory might not be strictly required, even if, for non-trivial tasks, it is often needed, especially when considering changing environments. Note that memory can be a stable structure, but it can also be alternative attractors.

The action policy has to be adjusted with respect to this feature and possibly other relevant features of the environment, such that the robot can reach its goals, in order for the robot to exploit a feature of the environment to improve its performance. Some questions arise as to what are these relevant features, what is the role of affordances, and how do policies emerge and improve. In addition, we also may ask the fundamental questions regarding the nature of computation: are policies calculated in an analogue calculation by a physical system? What is the character of the computation and improvements of policy? How is the policy carried out physically? For these latter questions, we can provide an answer by observing that from recent studies Hebbian and anti-Hebbian learning mechanisms have been discovered in biochemical networks of single-cell organisms [39]. These mechanisms involve protein translocation, signaling cascades, and chromatin memory, among others. In abstract terms, these systems can have dynamical attractors, can alter they attractors by synapses alterations, can evolve, and can store information in old and new attractors.

Changes in sensors, actuators, or the controller, in general, affect the phase space of the robot; therefore, it can move to the adjacent possible [40], i.e., the space of opportunities that can be reached starting from the actual condition. As a consequence, a policy update mechanism is needed to add new states and actions to the current policy. The new states and actions are new symbols associated to “meanings”. How new meanings appear and how they are linked to affordances is discussed in the next section.

## 4. Affordances, Constraints and Semantics

The possibility of developing sensors and actuators, and consistently adapting the behavior policy enable the organisms to identify affordances and create constraints that stabilize these features, thus expanding to the adjacent realm of possibilities. Affordances are continuously created and they can be exploited, constraints are created, and so forth.

We explain this view by means of an example of robots operating in the physical world. Let us take a setting in which robots have to sweep the floor. Suppose that there are bumps in the room and the locations of the bumps are correlated with the locations of the recharging sockets on the floor. If the robots could detect the bumps and use this correlation, they could get to the floor socket faster. Now, let us suppose that robot chassis is made of aluminium and that a robot, in the course of sweeping the floor, stumble by accident into a rigid obstacle, and gets a dent at exactly the right place to fit with floor bumps. This accident turns out to be beneficial to the robot, which can now reach the recharging sockets quickly. If the dent was the phenotypical result of a mutation, this robot would have an evolutionary advantage and its dent would be propagated, and possibly improved, in further generations. If this is not the case, then this accident is anyway giving a surviving advantage to the robot. The dent affords the robot to match the bump and so reach the recharging area faster. Now, the dent becomes a bump detector and "dent" and "bump" come to exist as new meanings. Therefore, not only exploiting an affordance opens up new possibilities to the robot, but it also creates meanings, i.e., it makes sense of physical, grounded properties of the world. Note that there is no complete description of the affordances of the world because affordances are relative to the purposes of the organism. For example, if I am a mouse and a cat is nearby, the distance to a rock under which I can hide as compared to the distance to the cat and my running speed is the relevant affordance: there is no external description, there is no God-like view. All of these views of organisms are situated and specific with respect to getting to exist in a non-ergodic universe.

Creating affordances is creating information, which is, in fact, semantic information [41], as opposed to Shannon-based information, which propagates already existing syntactic information. Semantic information characterizes correlations in the environment that are useful for a system as they carry significant knowledge. Affordances are carved out of a continuum, e.g., bumps and dents become new relevant variables in a formal representation of the world. After the relevant variables have co-created themselves, we “can name them” and new meanings come to exist. The presence or absence of a bump, the presence or absence of a dent can now be represented by variables. As a consequence, we can assert that semantics comes first and syntax comes later: first, the bump means something for me (good or bad), later it gets named and the corresponding symbol can be used. In digital computers the computations operate on the syntax (bits), but there is no semantics. The world is not a theorem. Theorems are operations on bits, i.e., symbols, and are not the world: the world is the bumps and dents. For these reasons, the notion of affordance is a key concept in biosemiotics, where it has been discussed and extended since its original formulation [37]. Particularly suitable to the perspective taken in this paper is the definition of affordance provided by Campbell et al. [42], who propose defining “affordances as potential semiotic resources that an organism enacts (detects, reads, uses, engages) to channel learning-as-choice in its environment”.

Let us now come back to the robotic example and imagine that another robot experiences a damage in one of its wheels, so that a bump appears and this bump somehow fits the dent of the other robot: now the second robot can detect the dent in the first robot and find the recharging station in an easier way just by following the first robot. Bumps and dents, and bumps and dents detectors evolve in the world, so we can collectively coordinate our behaviors to get what we need [21]. This becomes our construction of mutually consistent biosemiotic systems [22,43,44]. In the biosphere, we have evolved to mutually create mutually consistent affordances: the evolution of life is the evolution of myriads of meanings. Therefore, finding the principles governing the emergence of the organisms is the foundation of biosemiotics [45,46,47].

This creation of meanings through new emerging functions and dynamical patterns that turn out to be useful to some organisms is also at the roots of the symbol grounding problem [48], which concerns the way that symbols are intrinsically represented in systems. This problem has been thoroughly discussed inside the community of artificial intelligence and artificial life [49,50,51,52] and we believe that our perspective puts the problem in a more general context, also related to constructive biology [53]. Strictly connected to the grounding problem is the *frame problem*, that deals with how an embodied and situated system can represent and interact with the world it lives in [54,55,56]. Note that the viewpoint of evolving and maintaining mutually consistent meanings provides a unified way for dealing with both the grounding and frame problem.

The continuous identification of affordances and the construction of constraints change the phase space of the organisms and expand it to the adjacent possible. We remark that it is not possible to prestate new functions that emerge in this open condition. As stated before, the uses of a screwdriver are indefinite and so are its affordances. Let us take the case of the use of an engine block as a chassis for the tractor: it is also possible to use the engine block as a paper weight or to crack a coconut on one of its corners. These are alternative possible uses of the same physical object and in evolution none, one, some, or all of these uses may come to exist, for example, by Darwinian preadaptation. Because there is no deductive relation between the use of the engine block as a chassis and the use of the engine block to crack open coconuts, there can be no deductive theory of the evolution of biospheres. We are beyond deductively entailing laws: evolving biospheres are radically free [15,21]. Here, we remark that evolved systems are built of physical parts that have multiple causal features: some of them might be selected for performing a specific function useful to the organism. For example, we say that the heart pumps blood, because we identify, in this feature, a property functional to the survival of an animal and we discard other irrelevant features (e.g., the heart also produces sounds). Not only the emergence of these functions is non-deducible, but it is typical of a *bricolage* process, in which causal features of objects that turn out to be useful are exploited. Note that the segmentation of an organism into separated parts is often just a convenient simplification of our description [57]. Conversely, in engineered robotic systems, each part has its own identity and it is optimized for one specific function: here, the interactions among parts are precisely modeled and there is no space for affording new functions nor new emerging relations.

Concluding, the open evolution of organisms in the biosphere seems to set simple yet hard requirements for the emergence and open evolution of artificial organisms: embodiment, the capability of discerning what is good and bad, mechanisms for evolving sensors and actuators, and behavior policy open to changes.

## 5. Criticality

The organisms in the evolving biosphere are very likely to be critical, i.e., their dynamical regime is at the boundary between order and disorder [58,59]. This conjecture has found strong support in biology, neuroscience, as well as computer science [60,61], and it can be expressed as the combination of two statements:critical systems are more evolvable than systems in other dynamical conditions as they attain an optimal trade-off between mutational robustness (i.e., mutations moderately perturb the phenotype, without introducing dramatic changes) and phenotypic innovation (i.e., mutations can introduce significant novelty in the phenotypes); and,critical systems have advantages over ordered or disordered ones, because they optimally balance information storage, modification and transfer, and achieve the best trade-off between the repertoire of their possible actions and their reliability.

Therefore, the property of being critical is very likely to be found in Kantian wholes that identify and exploit affordances in their environment and, furthermore, they cooperate among each other to survive. If organisms are critical, their parts are not necessarily so. The advantage of being critical comes from the necessity of interacting with other systems in a changing and dynamic environment [24]. Anyway, being critical is an advantage in evolution and because it is easier to find an advantageous coupling with the environment [62]. Therefore, we expect to find Kantian wholes critical, but their parts may or may not be critical, depending on the evolutionary path that occurred and the way they are coupled with the other constituents. As a consequence, in a scenario with the evolution of artificial organisms, criticality should play an important role and systems should be designed in such a way that critical dynamical regimes are favored. Notably, dynamical models, such as Boolean networks, have been shown to maximize mutual information [63], basin entropy [64], and transfer entropy [65] when poised at the critical regime; both measures are correlated with the capability of discriminating percept categories and act accordingly. This property is crucial for evolving organisms, because as the number of sensors that come to exist increases and organisms use those sensors, the number of possible combinations of percepts increases exponentially in the number of sensors. For example, if the sensors return binary values and the number of sensors *N* increases in time, then the number of possible percept patterns increases as $2^{N}$. This is just a lower bound, as we are assuming binary percepts; organisms in Nature are rather analogical and this means that they are able to deal with a huge number of "worlds" they can sense.

## 6. Discussion and Conclusions

The ability of knowing “what’s good or bad for me”, the possibility of developing sensors and actuators and the capability of adapting their own behavior policy are properties that enable organisms to evolve in an ever expanding phase space. The astonishing evolution of the biosphere is beyond physics: critical co-evolving Kantian wholes develop by following paths for which there are no entailing laws. The question now arises as to what extent artificial systems can be built, such that they are endowed with the properties listed above and if these properties are sufficient for the emergence of artificial organisms.

Current advances in AI and robotics suggest a positive answer to the first part of the question [66]. Promising attempts to the online embodied evolution of robots have been proposed [67,68]. Soft robotics [69], and unconventional computing systems [70] may provide a viable approach to the evolution of sensors and actuators, along with self-improvement of behavior policies (which, of course, may greatly benefit from current machine learning and AI techniques).

Therefore, we can envision the availability of evolving hardware in the next future and observe the emergence of artificial organisms. Nevertheless, a profound mystery comes now into play: what is the role of consciousness in the evolution of the biosphere? In fact, we need constraints to choose and do, i.e., we need *qualia* as constraints on what we free will choose to do. In quantum mechanics, if physicists choose to measure the position of a particle they will find the position; if they choose to measure an interference pattern, an interference pattern will arise. This puzzle is that it seems as if the choice of the physicist of what to measure changes the world (either a particle or an interference pattern comes to exist). In an analogous way, we may argue that when we choose to perceive and use bumps, bumps come to exist [71].

Therefore, by setting up experiments with artificial systems in an open-ended evolution, we are looking for the boundary between what can be achieved through the properties that are listed above without consciousness, and what can be achieved with free will and qualia.

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
