# Peer review of "Emergence of Organisms"

_entropy, 2020, doi:10.3390/e22101163_

Round 1

Reviewer 1 Report

[Numbers below refer to rows in the manuscript.] 

(A)

(201–218, 251–307) On affordance. The concept has been widely analysed in recent research. According to this, affordance is certainly a (bio)semiotic phenomenon, which will be relevant to mention. E.g.:

Campbell, Cary; et al. 2019. Learning and knowing as semiosis: Extending the conceptual apparatus of semiotics. Sign Systems Studies 47(3/4): 352–381.

(B)

(202, 137, etc) Instead of speaking of "usefulness" (202, etc) or "good or bad" (37, 137, 283, 316, 350) or "beneficial or disadvantageous" (21) it would be more adequate to speak about the relevance for (metabolic, behavioral) coordination, of intracellular cooperation (152). "Good or bad" refer directly to the existence of norms, normativity, while the latter requires explanation (normativity is not discussed in the paper). Thus I suggest to use slightly more general formulation from which norms may follow, while not yet assumed. This would mean pointing to the existence of the situations of incompatibility and conflicts (between options) that cannot stay unresolved and thus the "protocell" has to make a choice. Whether it will be "bad" or "good" choice is already a matter of consequences; moreover, the value may not be binary ("bad" or "good") but have more gradations already in its primitive appearance. 

(C)

The paper speaks about the origin of organism, while the concept of organism is not defined. The definition-like formulation that is provided (150–152) is using the term "organism" and therefore is circular. An explicit definition of organism will be necessary.

(D)

A simple sensory-motor loop can be intracellular (i.e. not necessarily requiring the outside of the cell (organism). The discourse in this paper seems to assume that behavior (functions, purposes, goals) always require the outside, an environment, while this is obviously too restrictive. 

(E)

(159, 404) The name Jakob von Uexküll should be spelled using lowercase "v" in "von". In the bibliogrraphy, one should use Uexküll, J. von (not Von Uexküll, J.).

(F)

The problems discussed in the paper are deeply biosemiotic problems, or rather the problems of origin of semiosis. Thus the link to biosemiotics (briefly mentioned in the paper) could be even more explicitly emphasised.

(G) 

This is a very good paper.

Author Response

First of all we would like to thank to the reviewers for carefully reading the manuscript and providing very useful comments and suggestions, which helped us improving the paper and clarifying some points. We are also pleased that they have liked the paper.

In the following we explain how we changed the paper in response to the reviewers’ requests and suggestions. Please note that, in addition to the final amended manuscript, for ease of the reviewers we also provide a pdf file in which the relevant parts of the text that were changed have been highlighted as colored text (added text in blue and deleted text in red).

(A)

(201–218, 251–307) On affordance. The concept has been widely analysed in recent research. According to this, affordance is certainly a (bio)semiotic phenomenon, which will be relevant to mention. E.g.:

Campbell, Cary; et al. 2019. Learning and knowing as semiosis: Extending the conceptual apparatus of semiotics. Sign Systems Studies 47(3/4): 352–381.

We thank the reviewer for this suggestion. We added the suggested reference in section 4 of the paper (lines 208-302 of the revised version), along with the general definition of affordance proposed therein, which is particularly suited to our perspective.

(B)

(202, 137, etc) Instead of speaking of "usefulness" (202, etc) or "good or bad" (37, 137, 283, 316, 350) or "beneficial or disadvantageous" (21) it would be more adequate to speak about the relevance for (metabolic, behavioral) coordination, of intracellular cooperation (152). "Good or bad" refer directly to the existence of norms, normativity, while the latter requires explanation (normativity is not discussed in the paper). Thus I suggest to use slightly more general formulation from which norms may follow, while not yet assumed. This would mean pointing to the existence of the situations of incompatibility and conflicts (between options) that cannot stay unresolved and thus the "protocell" has to make a choice. Whether it will be "bad" or "good" choice is already a matter of consequences; moreover, the value may not be binary ("bad" or "good") but have more gradations already in its primitive appearance.

We agree with this observation. In fact, the point concerning the choice that the robot makes was taken from Ashby and it needed to be clarified. Accordingly, we revised the text in the point in which this notion is discussed, i.e. in Section 3.1. We think that this clarification is sufficient for a correct interpretation of the various expressions we have used in this respect throughout the paper.

(C)

The paper speaks about the origin of organism, while the concept of organism is not defined. The definition-like formulation that is provided (150–152) is using the term "organism" and therefore is circular. An explicit definition of organism will be necessary.

We changed the text in Section 3 (from line 156 of the revised version) so as to avoid circularity in the definition of organism. To support our definition we also added references concerning auto-catalytic systems.

(D)

A simple sensory-motor loop can be intracellular (i.e. not necessarily requiring the outside of the cell (organism). The discourse in this paper seems to assume that behavior (functions, purposes, goals) always require the outside, an environment, while this is obviously too restrictive.

We thank the reviewer for pointing us out this imprecision. We changed the text explaining that the notion of sensory-motor loop we mentioned is typical of robotics, while a more general concept can be provided. See lines 202-204 of the revised version.

(E)

(159, 404) The name Jakob von Uexküll should be spelled using lowercase "v" in "von". In the bibliogrraphy, one should use Uexküll, J. von (not Von Uexküll, J.).

We amended the text accordingly. As for the bibliographic reference, it seems that the bibliography style provided by the journal forces “von” before the surname; we have then asked the help of Entropy typesetting experts to solve this problem.

(F)

The problems discussed in the paper are deeply biosemiotic problems, or rather the problems of origin of semiosis. Thus the link to biosemiotics (briefly mentioned in the paper) could be even more explicitly emphasised.

We agree with the reviewer. Indeed, in our work we are at the foundations of biosemiotics. We further emphasized this link (see line 310-311 in the revised version) and added more references to biosemiotics.

(G)

This is a very good paper.

We thank the reviewer for this kind comment.

Reviewer 2 Report

This is a very important and timely essay on 'artificial life' that clearly posits the basic epistemological issues that make artificial systems basically different from organic life. The chosen perspective is the one of system-software engineering, thus a mainly functional-mechanistic one that by the way is the most diffuse in these times (the metaphor is considering organisms as a computational device with eventual emergent properties). The authors clearly explain the need of an 'active designer' to build such machines (at odds with natural organisms) by exploiting the concept of 'affordance' even in light of Kantian phylosophy. This is a very convincing and clear explanation of the 'fault' standing between real organisms and man-made systems. My only remark is the fact that the authors only superficially refer to the basic STRUCTURAL (that in turn becomes functional) divide between artifacts and biological complex systems that, even if still not alive (i.e. proteins), reside in the twilight zone between what we can understand in terms of 'parts' like sensors, actuatiors and so forth that are well within the realm of mechanics  (if..then rules are formally identical to mechanical devices like those present in an old clock)..and what we can not.

Basically it is the  identification of 'parts' singularly optimized for a task that lacks in organisms, this absence of 'proper independent parts' traces back to the 'synthesis' as opposed to 'mechanical design' as explained in the attached file (Biomedical Engineering Challenges (A chemical engineering perspective) Wiley (2018) Piemonte V., Basile A., Ito t. (editors) , Chapter 6). In my opinion such structural perspective should be more clearly presented in the manuscript in order to give more strength to the proposed arguments.

In any case the authors must be complimented for a very brilliant paper that surely deserves publication.

Author Response

First of all we would like to thank to the reviewers for carefully reading the manuscript and providing very useful comments and suggestions, which helped us improving the paper and clarifying some points. We are also pleased that they have liked the paper.

In the following we explain how we changed the paper in response to the reviewers’ requests and suggestions. Please note that, in addition to the final amended manuscript, for ease of the reviewers we also provide a pdf file in which the relevant parts of the text that were changed have been highlighted as colored text (added text in blue and deleted text in red).

This is a very important and timely essay on 'artificial life' that clearly posits the basic epistemological issues that make artificial systems basically different from organic life. The chosen perspective is the one of system-software engineering, thus a mainly functional-mechanistic one that by the way is the most diffuse in these times (the metaphor is considering organisms as a computational device with eventual emergent properties). The authors clearly explain the need of an 'active designer' to build such machines (at odds with natural organisms) by exploiting the concept of 'affordance' even in light of Kantian phylosophy. This is a very convincing and clear explanation of the 'fault' standing between real organisms and man-made systems. My only remark is the fact that the authors only superficially refer to the basic STRUCTURAL (that in turn becomes functional) divide between artifacts and biological complex systems that, even if still not alive (i.e. proteins), reside in the twilight zone between what we can understand in terms of 'parts' like sensors, actuators and so forth that are well within the realm of mechanics (if..then rules are formally identical to mechanical devices like those present in an old clock)..and what we can not.

Basically it is the identification of 'parts' singularly optimized for a task that lacks in organisms, this absence of 'proper independent parts' traces back to the 'synthesis' as opposed to 'mechanical design' as explained in the attached file (Biomedical Engineering Challenges (A chemical engineering perspective) Wiley (2018) Piemonte V., Basile A., Ito t. (editors) , Chapter 6). In my opinion such structural perspective should be more clearly presented in the manuscript in order to give more strength to the proposed arguments.

We thank the reviewer for this comment, which gave us the opportunity of clarifying this point and providing further literature to support to our perspective. We have added a brief discussion about this (see lines 330-340 of the revised version) and added the reference suggested.

In any case the authors must be complimented for a very brilliant paper that surely deserves publication.

We thank the reviewer for appreciating our work.
